# The Impact of Surgical Telementoring on Reducing the Complication Rate in Endoscopic Endonasal Surgery of the Skull Base

**DOI:** 10.3390/diagnostics14171874

**Published:** 2024-08-27

**Authors:** Janez Ravnik, Hojka Rowbottom, Carl H. Snyderman, Paul A. Gardner, Tomaž Šmigoc, Matic Glavan, Urška Kšela, Nenad Kljaić, Boštjan Lanišnik

**Affiliations:** 1Department of Neurosurgery, Maribor University Medical Centre, 2000 Maribor, Sloveniatomaz.smigoc@ukc-mb.si (T.Š.); 2Departments of Otolaryngology and Neurological Surgery, University of Pittsburgh School of Medicine, Pittsburgh, PA 15213, USA; 3Department of Neurological Surgery, University of Pittsburgh School of Medicine, Pittsburgh, PA 15213, USA; 4Department of Otorhinolaryngology, Head and Neck Surgery, Maribor University Medical Centre, 2000 Maribor, Slovenia; 5Department of Endocrinology and Diabetology, Maribor University Medical Centre, 2000 Maribor, Slovenia; 6Department of Ophthalmology, Maribor University Medical Centre, 2000 Maribor, Slovenia; nenad.kljaic@ukc-mb.si

**Keywords:** pituitary neoplasms, telementoring, endoscopic endonasal approach, outcomes

## Abstract

Background: Pituitary adenomas represent the most common pituitary disorder, with an estimated prevalence as high as 20%, and they can manifest with hormone hypersecretion or deficiency, neurological symptoms from mass effect, or incidental findings on imaging. Transsphenoidal surgery, performed either microscopically or endoscopically, allows for a better extent of resection while minimising the associated risk in comparison to the transcranial approach. Endoscopy allows for better visualisation and improvement in tumour resection with an improved working angle and less nasal morbidity, making it likely to become the preferred surgical treatment for pituitary neoplasms. The learning curve can be aided by telementoring. Methods: We retrospectively analysed the clinical records of 94 patients who underwent an endoscopic endonasal resection of a pituitary neoplasm between the years 2011 and 2023 at Maribor University Medical Centre in Slovenia. Remote surgical telementoring over 3 years assisted with the learning curve. Results: The proportion of complication-free patients significantly increased over the observed period (60% vs. 79%). A gradual but insignificant increase in the percentage of patients with improved endocrine function was observed. Patients’ vision improved significantly over the observed period. By gaining experience, the extent of gross total tumour resection increased insignificantly (67% vs. 79%). Conclusions: Telementoring for the endoscopic endonasal approach to pituitary neoplasms enables low-volume centres to achieve efficiency, decreasing rates of postoperative complications and increasing the extent of tumour resection.

## 1. Introduction

Over the last three decades, the endoscopic endonasal approach (EEA) has become the predominant approach for surgical treatment of different skull base pathologies, and is currently recognised as the standard for most pituitary adenomas [1,2,3]. EEA enables surgery in parasellar regions and sagittal and coronal plane corridors to the central skull base [1,4]. Microscopic transsphenoidal surgery had long been considered the “gold standard” in the surgical treatment of pituitary neoplasms, and at the beginning, the endoscope was used solely as an assisting tool to explore the sellar cavity for residual tumour tissue, but due to its wide panoramic, up-close visualisation, and recent developments of endoscopic instrumentation and techniques, EEA has gained popularity [4,5].

EEA is associated with a steep learning curve, which is necessary to increase the effectiveness of EEA and decrease the length of surgery [6,7,8]. A great deal of learning takes place in the operating theatre; however, sufficient time has to be spent in a dissection laboratory to acquire enough anatomical knowledge [1,9]. Additionally, EEA requires the adoption of new technology and instrumentation with a transition from three-dimensional to two-dimensional visualisation; however, newer three-dimensional endoscope technology has been developed to improve visualisation and understanding of the anatomy, which mirrors the view offered by a traditional microscope [1,10].

Telementoring, whereby an expert employs telecommunication technology to guide a less experienced learner from a remote location, provides a unique solution to increase quality and enhance access to surgical care in low-volume medical centres [11,12].

Collaboration between a neurosurgeon and an ear, nose, and throat (ENT) surgeon is crucial for successful patient management in the operating theatre, as well as outside [1,2]. The development of a skull base team that is efficient in performing EEA is a demanding task, even more-so in a low-volume centre; thus, telementoring by an experienced centre represents a possible solution to establish proficiency in EEA when on-site mentoring cannot be established [1,9,11].

The collaboration between neurosurgeons and ENT surgeons in endoscopic endonasal surgery at the Maribor University Medical Center (MUMC) in Slovenia started in 2009. The Maribor surgeons (JR and BL) were the first to establish a skull base team at this centre. In 2010, they attended the endoscopic skull base surgery course at the University of Pittsburgh Medical Center (UPMC), followed by a visit to the skull base centre at UPMC for an additional two weeks. A collaboration was established with the co-authors (CS and PG). The MUMC surgeons repeated their visit to the UPMC course and skull base department four years later. The collaboration between the four surgeons continued with regular consultations and an annual endoscopic skull base surgery course with lectures and hands-on cadaver dissections conducted in Maribor attended by the two surgeons from UPMC. The telementoring programme started in 2013 and finished in 2015.

Throughout the telementoring programme and thereafter, several adaptations and improvements were made. The extent of safe resection gradually improved and extended endonasal approaches for skull base pathology were introduced. Regarding pituitary tumours, a great effort was required to improve endocrinological results for non-secretory and secretory adenomas. The main goal was to preserve the endocrinological function of the remaining pituitary gland with the gross total resection of the pituitary adenoma. Important progress was achieved with skull base reconstruction with the introduction of the nasoseptal flap [13]. A multidisciplinary team comprising a neurosurgeon, otolaryngologist-head and neck surgeon (OHNS), endocrinologist, and ophthalmologist was established to provide better patient care with greater continuity of care and less duplication of services.

This article analyses the results and complications after successfully applying a three-year telementoring programme by evaluating the extent of surgery, endocrinological results, and rate of complications after pituitary tumour surgery.

## 2. Material and Methods

### 2.1. Study Design and Patient Data

#### 2.1.1. Study Design

We conducted a retrospective analysis of patients who underwent endoscopic endonasal transsphenoidal surgery performed at the MUMC between January 2011 and December 2023. A total of 94 patients were identified, of whom 93 regularly attended postoperative follow-up appointments. Patients included in the study analysis had to be older than 18 years. The observed timeframe was divided into 4 periods: from 2011 to 2014, 2015 to 2017, 2018 to 2020, and 2021 to 2023. For statistical comparison, the four groups of patients were further united into early (year 2011–2017) and late (year 2018–2023) periods.

During the telementoring programme, which started in 2013 and finished in 2015, 10 endoscopic endonasal surgeries of the skull base were mentored preoperatively and during the key part of the procedure. The telementoring programme required consent and a liability waiver from both groups, and the main team was solely responsible for the surgery and the outcome. Surgical instrumentation was standardised, and surgical techniques were adopted from the UPMC Center for Cranial Base Surgery. An educational programme was completed at the courses organised by the University of Pittsburgh Center for Cranial Base Surgery.

Prior to the start of telementoring, the technological needs and capabilities of both institutions were determined, and real-time two-way video and audio streaming was established using existing technology (Polycom Video Conferencing Equipment, Karl Storz Image 1 and Aida Video System, Tuttlingen, Germany). The surgical plan was laid out before surgery, and imaging was reviewed by both groups. Operative times were adjusted so that the key period of telementoring was in the range of 5:30–8:30 EST. Telementoring started and continued during the critical phases of the surgery, resection and reconstruction. Following each procedure, an evaluation form was used to document the interventions and rates of experience. Postoperative radiographs were shared via e-mail [1].

#### 2.1.2. Patient Data and Institutional Review Board

The local ethics committee of the MUMC reviewed and approved our study. Patient data were collected via the online medical record system, and the imaging examination data were collected via the local PACS system. The following data were collected from the electronic medical record: year of surgery, gender, age, length of hospitalisation, type of pituitary neoplasm and its location, exact cell type, postoperative complications, extent of resection, rehospitalisation after surgery for pituitary neoplasm, hypopituitarism before surgery, tumour hormonal activity, tumour size, hormonal syndrome before surgery, neurologic symptoms/signs before surgery, residual tumour tissue seen on control MRI 6 months after surgery, tumour recurrence after surgery and treatment, endocrine function before and after surgery, vision before and after surgery, and presence of hypopituitarism after surgery. Statistical data analysis was performed using the Statistical Package for Social Sciences (IBM SPSS Statistics for Windows, Version 29.0.2.0, Armonk, NY, USA). A Pearson Chi-Square test and Fisher’s exact test for 2 × 2 tables were used for the statistical comparison of the groups’ frequencies. *p* value < 0.05 was considered statistically significant.

#### 2.1.3. Clinical Management

Each patient with a sellar pathology was presented to the multidisciplinary team, which examined the MRI imaging, the results of the endocrinologic tests, and the results of the neuro-ophthalmic examination. Patients with visual deficits, growing neoplasms, and uncontrollable hormonal deficits were identified as suitable candidates for surgery. The operative procedure, purpose, and risks were explained to every patient who chose operative treatment or conservative management. Most patients were hospitalised at the University Department of Otorhinolaryngology, Head and Neck Surgery, and a few at the Department of Neurosurgery. The surgeries were performed in a dedicated endoscopic operating room enabling a four-handed endoscopic technique. Image guidance was used in all cases. The telementoring link was established and tested prior to the surgery. Each surgery started with the preparation of the nasoseptal flap, usually on the right side. A one-sided corridor was created, and bilateral sphenoidotomy was performed. At this stage, telementoring started. The most common intervention from the overseeing team was increased exposure at the level of the rostrum and medial opticocarotid recesses (medial OCR) to improve suprasellar exposure. Particular care was taken to identify both carotid and optic protuberances. The sellar floor was removed in the usual fashion. At this point, the performing surgeons were usually advised to remove more bone in all directions to reach optic and carotid protuberances and gain sufficient access to the intrasellar pathology. The resection phase was also subject to telementoring, with tips on tumour resection in the suprasellar region. The performing surgeons were instructed to carefully inspect the boundaries of the tumour, and they were assisted in properly identifying tumour borders and the remnant of the pituitary gland. Special care was taken not to damage the healthy pituitary tissue. The removal of the tumour tissue with two aspirators (one with a neuronavagated tip) was extremely useful. Reconstruction was monitored only in the first intradural phase. After ensuring watertight intradural reconstruction, the defect was covered with a nasoseptal flap. After surgery, patients were transferred to the intensive unit within the Department of Otorhinolaryngology while awake. Packing was removed between the 3rd and 5th postoperative day.

After discharge, they were examined by an endocrinologist 3 months following surgery, where endocrinologic tests were repeated and hormonal replacement therapy was adjusted. After surgery, all patients had the following follow-up appointments: 3 months after surgery, they were re-examined by an endocrinologist; and 6 months after surgery, a control MRI was performed, which was examined by two independent neuroradiologists to assess the extent of tumour resection, and they were later examined by a neurosurgeon and an ophthalmologist. Patients with no radiologic and/or endocrinologic signs of tumour recurrence were followed-up with an annual MRI and regular appointments with an endocrinologist, neurosurgeon, and ophthalmologist. Patients with a recurrence of sellar pathology were either further treated or monitored.

## 3. Results

From 2011 to 2023, 94 patients underwent endoscopic endonasal surgery for a sellar pathology, and from that sample, 49 (52.1%) were women. The mean age of the included patients was 58 years (range 20–82 years, SD = 14).

Table 1 demonstrates the number and proportion of operated patients per observed period. During the telementoring programme (2013–2015), ten endoscopic endonasal surgeries of the skull base were performed, and clinical follow-up was provided with postoperative imaging diagnostics shared by e-mail between the consultants with intense communication. No technical difficulties were encountered during the telementoring programme with a good and stable audio-visual connection. The general results of the telementoring programme were described in the previous publication [1].

The operations performed from 2011 to 2012 were conducted with no help of telementoring from an experienced centre, and surgeries after 2015, when the telementoring period finished, comprised discussions amongst surgeons; however, these were not live during surgery, but were either prior to or after the operations.

Adenomas represented the most common pituitary neoplasm in the observed group, with null cell adenomas diagnosed in 45.7% of the observed population, plurihormonal adenomas present in 31.9%, and monohormonal adenomas in 13.8% of patients.

The majority of tumours were in the intrasellar and suprasellar regions. Most patients were diagnosed with macroadenomas, and more than half had hypopituitarism before surgery. The hormonal activity of the pituitary neoplasm was present in 20.2% of cases, and from those, more than half (55.6%) were diagnosed with a growth-hormone-secreting adenoma, followed by prolactinomas in 38.9%, adrenocorticotropic hormone-secreting tumours in 11.1%, and thyroid-stimulating hormone secretion in 5.6% of cases, as illustrated in Table 2.

Around a quarter (25.5%) were diagnosed with a hormonal syndrome before surgery, with hyperprolactinemia being the most common one, followed by acromegaly, and 84% had neurological symptoms and deficits, with vision impairments being the most common ones, as illustrated in Table 3.

There was one death in our series of patients, as demonstrated in Table 4 and Figure 1. The cause of death was a respiratory infection unrelated to the surgical procedure. Over the observed period, the percentage of complication-free patients after surgery gradually increased, as is demonstrated in Table 4. Diabetes insipidus was transient in all cases. In the early period (from 2011 to 2017), 19 patients were complication-free, whereas in the late period (from 2018 to 2023), 45 were complication-free. The difference in the overall number of complications is statistically significant between the early and late period (*p* = 0.022, Fisher’s Exact Test). However, the difference between the two periods (early vs. late) for each individual complication was not statistically significant (*p* > 0.05, Fisher’s Exact Test).

A gradual increase in improved endocrine function after surgery, which was determined by the reduction in hormonal replacement therapy or its discontinuation, was observed, as well as an improvement in patients’ vision after surgery, as is shown in Table 5 and Figure 2. When comparing postoperative endocrine function and postoperative hypopituitarism, there were no statistically significant differences between the early (2011–2017) and late period (2018–2023) (*p* > 0.05, Fisher’s Exact Test). However, patients’ eyesight improved significantly (*p* = 0.013, Fisher’s Exact Test) in the late period.

The extent of neoplasm resection, which was ascertained using MRI 6 months after surgery, progressively increased from 66.7% in the first period to 79.4% in the last period, as illustrated in Table 6 and Figure 3; however, a chi-square test of independence showed that the difference was not statistically significant (X^2^ (3, N = 94) = 3.788, *p* = 0.285).

The proportion of patients who required rehospitalisation after EEA gradually decreased. After being discharged, four patients (26.7%) were rehospitalised in the first period; one was diagnosed with a nasal septal abscess and was discharged after 4 days; the second patient was diagnosed with diabetes insipidus and was discharged after 5 days; and two patients had suboptimal hydrocortisone therapy and were both discharged after 2 days. In the second period, one patient (4.8%) had to be rehospitalised for an additional 40 days, as they developed a cerebrospinal fluid leak and meningitis. In the third period, one patient (4.2%) was re-admitted for 12 days for additional tumour resection after the control MRI; there were no postoperative complications.

Out of 94 patients, 11 had recurrences of the following: Rathke cleft cysts and adenoma that required no revision surgery each recurred in 1 patient; a craniopharyngioma that required additional surgery was present in 1 patient; and 8 patients had adenomas that regrew and required revision surgery. EEA was utilised in all cases of revision surgery.

The senior surgeons at Maribor operated on 22 patients before becoming proficient enough to operate without telementoring from surgeons at UPMC. Pituitary tumours present the ideal first step in the endoscopic learning curve based on their location in the centre of the skull base outside the subarachnoid space and with the natural corridor provided by the sphenoid sinus in line with the nasal cavity [14].

## 4. Discussion

This study demonstrates a successful implementation of EEA for pituitary neoplasms in a low-volume centre, such as MUMC, with an emphasis on the combined efforts of a newly established skull base team and their collaboration with an experienced skull base centre. The number of patients gradually increased over the observed period, with 34 patients having been operated on in the last 3 year period from 2021 to 2023 in comparison to only 15 in the first 4 year period from 2011 to 2014. The major reason for the rise in patient numbers was an increased referrals of patients from other hospitals to our institution. The steady increase in the number of patients operated on each year for pituitary tumours with the addition of other endoscopic endonasal procedures ensured a safe and efficient EEA in a low-volume centre.

This study has potential limitations. Despite a gradual increase in the number of operated patients, the number of patients included in the study was relatively small, since MUMC is a midsize hospital in a country with a population of approximately 2 million. Additionally, in this study, we included patients with different sellar pathologies, which share distinctive behaviours and characteristics; however, the main aim of our work was to demonstrate the acquirement of the surgical technique and the learning curve of the newly established skull base team with the help of telementoring from an experienced centre. Furthermore, the follow-up period for patients operated on in the last period (2021–2023) was considerably shorter than for those in the first period (2011–2014); hence, the comparison of long-term effects of treatment could not be performed.

Pituitary adenomas, benign neoplasms that comprise up to 15% of all intracranial masses, represent the most common pituitary disorder, with an estimated prevalence as high as 20% based on autopsy and radiologic studies, with the majority having no clinical significance [14,15,16,17,18]. Neoplasms can manifest with syndromes of hormone hypersecretion or deficiency, neurological symptoms from a mass effect of the expanding gland, or they can be an incidental finding on imaging performed for unrelated issues [19,20,21]. In our observed patient population, the neurological signs and symptoms were the most common, present in more than 80% of cases. Treatment aims to reduce hormone hypersecretion, remove the mass effect, and correct hormone deficiency [22,23,24]. Nonfunctioning asymptomatic microadenomas do not require immediate treatment; however, patients require careful monitoring with regular control MRIs [25,26].

Transsphenoidal surgery for the resection of pituitary neoplasms can be performed either microscopically or endoscopically, and has been proven to achieve a better extent of resection while minimising the associated risks in comparison to the transcranial approach; thus, it represents the first-line treatment for the vast majority of pituitary adenomas [14,27,28]. Microscopic transsphenoidal surgery used to be the gold standard for pituitary tumour removal; however, poor visualisation due to the deep location of adenomas, combined with a narrow surgical space, presented a major obstacle [4]. Endoscopy, with its panoramic view, has led to better visualisation and, thus, improvements in the extent of tumour resection; it also provides an improved working angle and less nasal morbidity [28,29,30]. The growing acceptance of endoscopic transsphenoidal surgery for pituitary adenomas amongst surgeons is making it highly likely to become the first line of approach for pituitary neoplasm in the future [27,31,32,33]. Currently, at the MUMC, with the establishment of a proficient skull base team, EEA represents the main surgical modality in the treatment of pituitary pathologies.

EEA allows for surgery in the parasellar regions, namely planum/tuberculum sellae, cavernous sinus, and clivus, where larger and more complex adenomas invade, as the endoscope can be introduced through narrow corridors to reach deep spaces, and with angled lenses, it provides enhanced vision to achieve what is commonly known as “looking around the corners” [4,14].

Endocrinologic outcomes after endoscopic adenoma removal have been comparable, and in some cases even superior, to those after microsurgical tumour removal [34]. A systematic review by Strychowsky demonstrated that EEA for pituitary neoplasms is associated with less mean blood loss, shorter hospitalisation and operative time, fewer nasal complications, and an increase in the proportion of gross total tumour resection with decreasing incidence of postoperative diabetes insipidus [35].

The learning curve for EEA to the pituitary tumour is steep and is often associated with an increase in complications, as the surgeon is faced with unfamiliar anatomy and new technologies, as well as new surgical skills [9,28]. The ideal number of cases to achieve proficiency is not established and depends on the surgeon’s prior experiences and skills, the composition of the practice, and the frequency of surgeries [36]. Based on limited published data, it is recommended that a team should perform 30 to 50 pituitary surgeries together to gain proficiency prior to taking on more difficult surgical cases, namely EEA to vascular malformations and highly vascular tumours [9]. The surgeons at MUMC performed 22 operations before becoming accomplished; however, they had previous experience with other endonasal endoscopic procedures that they could build on. Due to its steep learning curve, mentorship is vital for mastering the EEA; however, centralisation of care in high-income countries can restrict access to surgical training outside of metropolitan hubs. Telementoring can help face these challenges, as it can be used effectively in a variety of settings [12,37,38]. Studies have shown that there was no significant difference in outcomes for trainees who received telementoring or on-site mentoring [11]. The advent of augmented and virtual reality utilised for telementoring could further increase autonomy for trainee surgeons [39,40,41]. The main disadvantages to a wider implementation of telementoring are costs, technical limitations, reliable internet connection, and cyber security [11,12]. The potential drawbacks of telementoring must be addressed before the programme starts. Both groups involved must be familiar with surgical hands-on capabilities and dexterity, and mutual trust is paramount for success. There is also a question of liability that must be cleared up before the surgery. In our case, the domestic team at MUMC was solely responsible for the management of patients as well as potential complications. A team must have a surgical plan laid out before the surgery, as well as a backup plan in case of problems with audio-visual connection. Despite some potential limitations, the benefits of telementoring outweigh them if the programme starts under careful consideration and in a collaborative environment.

The primary goal of most pituitary surgeries is gross total resection. In a study conducted by Dehdashti, gross total resection of macroadenomas was achieved in 96% of cases [35]. Similarly, Wang and colleagues managed to achieve a gross total resection in 92% of cases, whereas total resection was achieved in 76% of cases of giant adenomas [24]. Yu and colleagues cited a gross total resection of pituitary adenomas to be achieved in 60–73% of all cases, which was higher than expected based on a systematic review conducted by Komotar, where gross total resection was achieved in 47.2% of cases [30,42,43,44]. In our patient series, the proportion of cases of GTR increased from 66.7% in the first observed period up to almost 80% (79.4%) in the last period; however, the difference was not statistically significant when comparing the extent of resection in the early period (2011–2017) and the late one (2018–2023). In the first period, subtotal and partial resections were present more often than in other periods, as the primary focus was avoiding postoperative complications rather than the extent of resection. Wang and colleagues managed to achieve the GTR of macroadenomas in the majority of cases by using intraoperative MRI and neuronavigation during surgery for assessing residual tumour tissue and determining further safe tumour resection, second transcranial surgery, radiotherapy, or radiosurgery [30]. The usefulness of intraoperative MRIs is controversial, with some studies reporting a higher rate of gross total resection, but others showing no difference [45,46].

Patients with nonfunctioning pituitary adenomas most often present with signs and symptoms of hypopituitarism, headaches, hemianopsia, progressive loss of vision, and diplopia [30,45,47,48,49,50,51,52]. Headaches are present in 16% to 70% of cases, and are mainly localised in the frontal and occipital regions [45,53,54]; in our patients series, 36.2% presented with headaches. Visual impairments, most often bitemporal visual deficits due to mid-chiasmal compression, are present in larger adenomas, whereas diplopia is rare [3,55,56]. Amongst our patient group, 70.2% had visual deficits. Additionally, nonfunctioning pituitary adenomas, especially macroadenomas, can cause hypopituitarism with at least one hormone deficiency [57,58]. The patients in our study were mainly diagnosed with macroadenomas (97.9%), and 58.5% had hypopituitarism before surgery. Asymptomatic nonfunctioning adenomas are not recommended for surgical treatment, except for young patients. In cases of incomplete tumour resection with no obvious symptoms, patients are recommended for further observation rather than second surgery, which is required in cases of tumour regrowth [30,59]. Following surgery for nonfunctioning adenomas, visual impairments often improve, whereas data regarding hypopituitarism is inconsistent [60,61]. In our case series, the proportion of patients whose vision improved after pituitary tumour removal increased over the observed period from 26.7% in the first period to 64.7% by the last period. When comparing patients’ postoperative vision between the early period (2011–2017) and the late one, there was a statistically significant improvement. An improvement in postoperative endocrine function was also observed over the period; in the first period, 6.7% experienced endocrine function improvement, whereas by the last period, that proportion increased to 17.6%; however, it was not statistically significant.

In a study conducted by Alexopoulou, 80% of patients with nonfunctioning adenomas had at least one pituitary axis deficiency, and following surgery, that proportion dropped to 61%, with the improvement mostly present in cases of adenomas affecting the LH/FSH and TSH axis [62]. Hypopituitarism was present in 58.5% of cases before surgery and 68.1% after surgery in our series; however, over the observed period, we managed to reduce the percentage of cases with hypopituitarism following surgery from 86.7% in the first period to 52.9% in the last. With experience, distinguishing the normal pituitary gland from the tumour tissue was easier.

Tumour size is recognised as a preoperative predictor of new pituitary deficiency, with surgery of larger neoplasms more often leading to a new hormone deficiency [60]. The function of the hypothalamic-pituitary axis continues to change postoperatively, even without radiotherapy [60]. Surgery for prolactinomas is suggested when patients are intolerant to the side effects of medications or they are ineffective, and in cases of pituitary apoplexy [30,63].

Complications after endoscopic surgery for pituitary adenomas occur in the range from 3.4% to 36.1% with a 1% mortality [18,30,45,64,65,66]. In our patient series, the percentage of complications following surgery gradually decreased over the observed period from 40% in the first period to 20.6% in the final period. The most often encountered postoperative complications are diabetes insipidus, anterior lobe dysfunction, and cerebrospinal fluid leak [30]. In our study, 16% developed diabetes insipidus after surgery, which was transient in all cases, cerebrospinal fluid leak was present in 6.4%, and intracranial haemorrhages that required revision surgery occurred in 7.4%.

Most cases of cerebrospinal fluid leak stop spontaneously during surgery; however, their incidence is higher in cases of macroadenomas, and a tear in the diaphragm or arachnoid membrane has to be appropriately reconstructed at the end of the procedure [18,64,67,68]. The risk of cerebrospinal fluid leak after EEA is increased in larger adenomas with suprasellar extension, intraoperative leakage, repeat surgery, and high body mass index [69,70]. The rate of cerebrospinal fluid leak after transsphenoidal surgery for pituitary adenomas is approximately 5%, with no significant difference between the endoscopic and microscopic approach [14,71,72]. Over the observed period, the percentage of patients who required rehospitalisation after surgery also decreased from 26.7% in the first period to 2.9% in the fourth period. By effectively reconstructing the sellar floor and utilising the multilayer techniques and nasoseptal grafts, we managed to decrease the complication rates following EEA for pituitary tumours.

No clear evidence is available on the timing, frequency, and duration of postoperative endocrine, radiologic, and ophthalmologic assessments, with most studies suggesting postoperative endocrine evaluation 4 to 8 weeks after surgery and others suggesting 2 to 6 months postoperatively [45,73]. MRI imaging performed immediately after surgery can be misleading due to debris, blood, and packing material; therefore, it is usually performed 3 to 6 months after the operation, when most postoperative changes cease [45,52,74]. In our study, the control MRI was performed 6 months after surgery. The interval for further MRI follow-up is decided upon residual tumour size and its distance to the optic chiasm [45]. Postoperatively, 68% of patients with preoperative visual impairment experience an overall improvement and approximately 5% deteriorate, with longer duration of visual field deficits and severity of visual symptoms being linked to worse postoperative visual outcomes [56,75,76,77]. Overall, in our study group, 2.9% of patients experienced a decline in their postoperative vision. It has been suggested that a visual examination be performed 3 months after surgery and then from every 4 to 6 months until stabilisation of the visual function, since visual defects tend to progressively improve, especially in the first year following surgery [78,79]. Nonfunctioning pituitary adenomas have regrowth rates between 15% and 66% when treated with surgery or 2% to 28% when combining surgery with radiotherapy; therefore, long-term follow-up is recommended [80,81]. Tumour recurrence was present in 11.7% of cases in our study, with a majority (81.8%) being adenomas, and most recurrences (81.8%) were treated using a second operation, and a minority were monitored using regular control MRI (18.2%).

The recurrence rate peaks between 1 and 5 years after surgery and declines after 10 years; hence, 10 or more years of postoperative imagining surveillance is indicated, with some suggesting lifelong monitoring of patients after surgery for pituitary adenoma, especially in cases of tumour remnants [45,81]. Radiosurgery was utilised in 3.2% of cases in our study with no further recurrence; one was a case of adenoma and two were metastases.

A grading system based on predicting factors, such as tumour invasion seen on MRI, immunohistochemical profile, mitotic index, and Ki-67 and p53 positivity, has been recently suggested to identify patients with a high risk of recurrence of progression [82]. Combining surgery and radiotherapy has been more effective than surgery alone in tumour recurrence prevention; however, radiotherapy can cause significant side effects, namely radiation-induced optic neuropathy, hypopituitarism, and secondary brain tumours; thus, it is reserved for cases of incomplete resection of adenomas with high proliferative activity or in cases of recurrence after repeated operations [83,84].

## 5. Conclusions

EEA represents a safe and effective modality for the treatment of pituitary adenomas where GTR is vital. Telementoring from an experienced centre enables less-experienced surgeons to establish a proficient skull base team with improving results, and it represents a cost-effective model for the global education of surgeons.

## Figures and Tables

**Figure 1 diagnostics-14-01874-f001:**
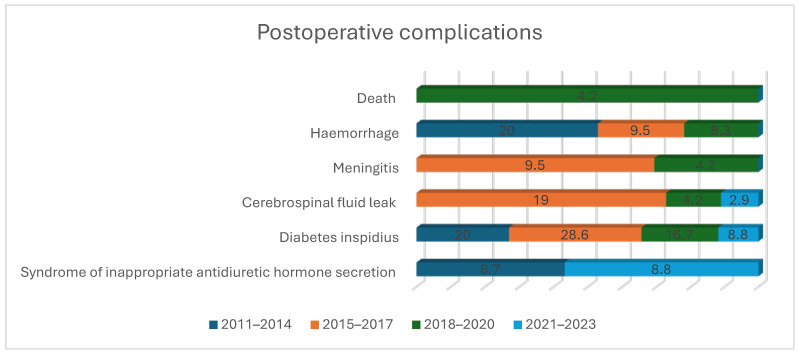
Postoperative complications per observed period.

**Figure 2 diagnostics-14-01874-f002:**
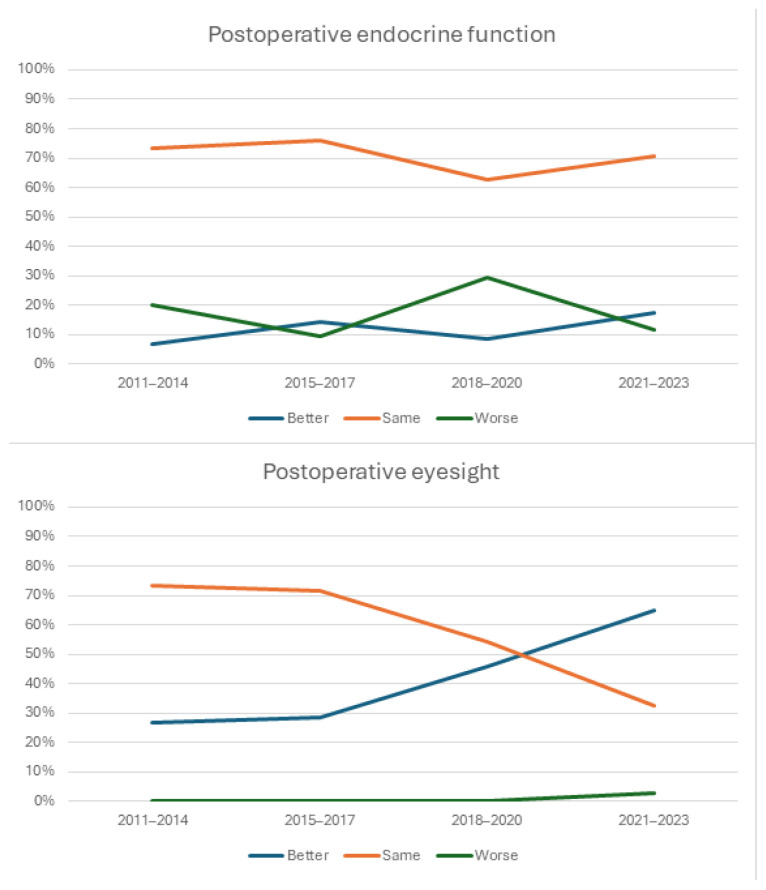
Postoperative endocrine function and eyesight per observed period.

**Figure 3 diagnostics-14-01874-f003:**
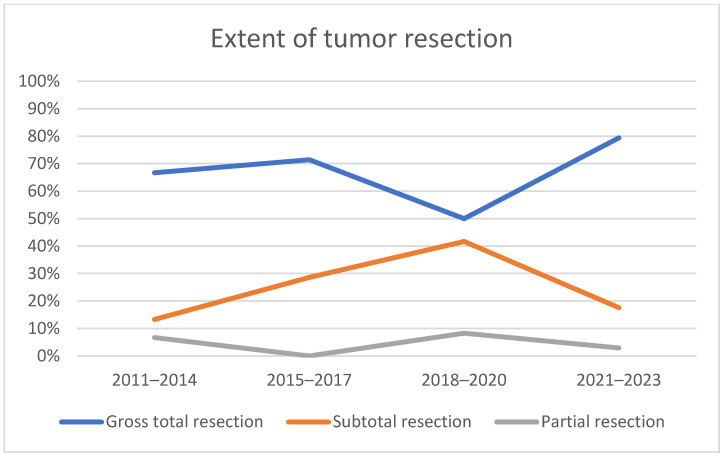
Extent of tumour resection per observed period.

**Table 1 diagnostics-14-01874-t001:** Number and percentage of patients operated per period.

		Number	Percentage
Period	2011–2014	15	16.0%
	2015–2017	21	22.3%
	2018–2020	24	25.5%
	2021–2023	34	36.2%
	Total	94	100.0%

**Table 2 diagnostics-14-01874-t002:** Sellar pathology in the observed group of patients.

		Number	Percentage
Neoplasm type	Adenoma	86	91.5%
	Craniopharyngioma	2	2.1%
	Schwannoma	1	1.1%
	Metastasis	3	3.2%
	Rathke cleft cyst	1	1.1%
	Lymphocytic hypophysitis	1	1.1%
	Total	94	100.0%
Neoplasm location	Intrasellar region	16	17.0%
	Intrasellar and suprasellar region	72	76.6%
	Intrasellar, suprasellar and parasellar region	4	4.3%
	Intrasellar and parasellar region	2	2.1%
	Total	94	100.0%
Hypopituitarism prior surgery	No deficit	39	41.5%
	Hypopituitarism	55	58.5%
	Total	94	100.0%
Hormonal activity	No	75	79.8%
	Yes	19	20.2%
	Total	94	100.0%
Size	Microadenoma	2	2.1%
	Macroadenoma	92	97.9%
	Total	94	100.0%

**Table 3 diagnostics-14-01874-t003:** Preoperative signs and symptoms.

		Number	Percentage
Cushing syndrome	No	90	95.7%
	Yes	4	4.3%
	Total	94	100.0%
Hyperprolactinemia	No	82	87.2%
	Yes	12	12.8%
	Total	94	100.0%
Hyperthyroidism	No	93	98.9%
	Yes	1	1.1%
	Total	94	100.0%
Acromegaly	No	84	89.4%
	Yes	10	10.6%
	Total	94	100.0%
Vision deficits	No	28	29.8%
	Yes	66	70.2%
	Total	94	100.0%
Apoplexy	No	81	86.2%
	Yes	13	13.8%
	Total	94	100.0%
Seizures	No	93	98.9%
	Yes	1	1.1%
	Total	94	100.0%
Headaches	No	60	63.8%
	Yes	34	36.2%
	Total	94	100%

**Table 4 diagnostics-14-01874-t004:** Occurrence of postoperative complications per observed period.

		Period							
		2011–2014		2015–2017		2018–2020		2021–2023	
		Number	Percentage	Number	Percentage	Number	Percentage	Number	Percentage
Syndrome of inappropriate antidiuretic hormone secretion	No	14	93.3%	21	100.0%	24	100.0%	31	91.2%
	Yes	1	6.7%	0	0.0%	0	0.0%	3	8.8%
	Total	15	100.0%	21	100.0%	24	100.0%	34	100.0%
Diabetes insipidus	No	12	80.0%	15	71.4%	20	83.3%	31	91.2%
	Yes	3	20.0%	6	28.6%	4	16.7%	3	8.8%
	Total	15	100.0%	21	100.0%	24	100.0%	34	100.0%
Cerebrospinal fluid leak	No	15	100.0%	17	81.0%	23	95.8%	33	97.1%
	Yes	0	0.0%	4	19.0%	1	4.2%	1	2.9%
	Total	15	100.0%	21	100.0%	24	100.0%	34	100.0%
Meningitis	No	15	100.0%	19	90.5%	23	95.8%	34	100.0%
	Yes	0	0.0%	2	9.5%	1	4.2%	0	0.0%
	Total	15	100.0%	21	100.0%	24	100.0%	34	100.0%
Haemorrhage	No	12	80.0%	19	90.5%	22	91.7%	34	100.0%
	Yes	3	20.0%	2	9.5%	2	8.3%	0	0.0%
	Total	15	100.0%	21	100.0%	24	100.0%	34	100.0%
Death	No	15	100.0%	21	100.0%	23	95.8%	34	100.0%
	Yes	0	0.0%	0	0.0%	1	4.2%	0	0.0%
	Total	15	100.0%	21	100.0%	24	100.0%	34	100.0%
Other complications	No	15	100.0%	20	95.2%	23	95.8%	34	100.0%
	Yes	0	0.0%	1	4.8%	1	4.2%	0	0.0%
	Total	15	100.0%	21	100.0%	24	100.0%	34	100.0%

**Table 5 diagnostics-14-01874-t005:** Postoperative endocrine function and eyesight per observed period.

		Period							
		2011–2014		2015–2017		2018–2020		2021–2023	
		Number	Percentage	Number	Percentage	Number	Percentage	Number	Percentage
Endocrine function	Better	1	6.7%	3	14.3%	2	8.3%	6	17.6%
	Same	11	73.3%	16	76.2%	15	62.5%	24	70.6%
	Worse	3	20.0%	2	9.5%	7	29.2%	4	11.8%
	Total	15	100.0%	21	100.0%	24	100.0%	34	100.0%
Eyesight	Better	4	26.7%	6	28.6%	11	45.8%	22	64.7%
	Same	11	73.3%	15	71.4%	13	54.2%	11	32.4%
	Worse	0	0.0%	0	0.0%	0	0.0%	1	2.9%
	Total	15	100.0%	21	100.0%	24	100.0%	34	100.0%
Hypopituitarism after surgery	No	2	13.3%	7	33.3%	5	20.8%	16	47.1%
	Yes	13	86.7%	14	66.7%	19	79.2%	18	52.9%
	Total	15	100.0%	21	100.0%	24	100.0%	34	100.0%

**Table 6 diagnostics-14-01874-t006:** Extent of tumour resection per observed period.

		Period							
		2011–2014		2015–2017		2018–2020		2021–2023	
		Number	Percentage	Number	Percentage	Number	Percentage	Number	Percentage
Extent of tumour resection	Gross total resection	10	66.7%	15	71.4%	12	50.0%	27	79.4%
	Subtotal resection	2	13.3%	6	28.6%	10	41.7%	6	17.6%
	Partial resection	1	6.7%	0	0.0%	2	8.3%	1	2.9%
	Biopsy	2	13.3%	0	0.0%	0	0.0%	0	0.0%
	Total	15	100.0%	21	100.0%	24	100.0%	34	100.0%

## Data Availability

The original contributions presented in the study are included in the article, further inquiries can be directed to the corresponding author.

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
