# Peer review of "The Impact of Surgical Telementoring on Reducing the Complication Rate in Endoscopic Endonasal Surgery of the Skull Base"

_diagnostics, 2024, doi:10.3390/diagnostics14171874_

Round 1

Reviewer 1 Report

Comments and Suggestions for Authors

The authors investigated the impact of surgical telementoring in reducing complication rate in endoscopic endonasal surgery of the skull base. I have the following comments:

1.      Abstract: OK.

2.      Introduction: OK.

3.      Materials and methods: The full term of SPSS should be provided.

4.      Results:

a.       Please add the SD to the mean.

b.      Please move this "The observed timeframe was divided into 4 periods: from 2011-2014, 2015-2017, 2018-2020, and 2021-2023. For statistical comparison, the four groups of patients were further united into early (year 2011-2017) and late (year 2018-2023) period." to the methodology section.

c.       Table 1: Ethier put the total patients' number (94) in the legend or in the table itself.

d.      Table 2: Please see the above-mentioned comment. Besides, you must complete the last three variables to be fit with the total number of patients.

e.       Line 156: This percentage 20.9% should be the same as in Table 2.

f.        Lines 159-161: This "Table 2 demonstrates the different neoplasms and their locations, neoplasm size, and presence of hypopituitarism before surgery as well as hormonal activity of the neoplasm." Is a repetition. You can refer your results to Table 2 only.

g.      Table 3:

1.      You should add the patients' number (94) to its legend.

2.      Please delete the word "column" from the table.

3.      Please rearrange the table to be fit with text.

h.      Table 4:

1.      You should add the patients' number (94) to its legend.

2.      Please delete the word "column" from the table.

3.      Please write the full term of the abbreviations.

4.      You must add the statistical test and P-values in the table.

i.        Tables 5 and 6: Please see the comments on Table 4.

5.      Discussion: Please add any limitations to the study.

6.      Out of 77 references, only 11 belongs to the last five years (2019-2023), therefore it is necessary to update the references. No reference belongs to the year 2024. This link https://pubmed.ncbi.nlm.nih.gov/?term=Endoscopic+endonasal+excision+of+pituitary+adenoma&filter=years.2024-2024 might be useful for your study.

Comments on the Quality of English Language

The article needs minor editing.

Author Response

Response to Reviewer 1

The authors investigated the impact of surgical telementoring in reducing the complication rate in endoscopic endonasal surgery of the skull base. I have the following comments:

  1. Abstract: OK.
  2. Introduction: OK.
  3. Materials and methods: The full term of SPSS should be provided.

Response: Thank you very much for your comment. We have made revisions accordingly.

  1. Results:
  2. Please add the SD to the mean.

Response: Thank you very much for your comment. We have made revisions accordingly.

  1. Please move this "The observed timeframe was divided into 4 periods: from 2011-2014, 2015-2017, 2018-2020, and 2021-2023. For statistical comparison, the four groups of patients were further united into early (year 2011-2017) and late (year 2018-2023) period" to the methodology section.

Response: Thank you very much for your comment. We moved the mentioned section from the Results to the Methods.

  1. Table 1: Ethier put the total patients' number (94) in the legend or in the table itself.

Response: Thank you very much for your comment. We have made revisions accordingly.

  1. Table 2: Please see the above-mentioned comment. Besides, you must complete the last three variables to be fit with the total number of patients.

Response: Thank you very much for your comment. We have made revisions accordingly.

  1. Line 156: This percentage 20.9% should be the same as in Table 2.

Response: Thank you very much for your comment. We have made revisions accordingly.

  1. Lines 159-161: This "Table 2 demonstrates the different neoplasms and their locations, neoplasm size, and presence of hypopituitarism before surgery as well as hormonal activity of the neoplasm." Is a repetition. You can refer your results to Table 2 only.

Response: Thank you very much for your comment. We have deleted the repetition in the text, which now flows better.

  1. Table 3:
  2. You should add the patients' number (94) to its legend.
  3. Please delete the word "column" from the table.
  4. Please rearrange the table to be fit with text.

Response: Thank you very much for your comment. We have made revisions accordingly.

  1. Table 4:
  2. You should add the patients' number (94) to its legend.
  3. Please delete the word "column" from the table.
  4. Please write the full term of the abbreviations.
  5. You must add the statistical test and P-values in the table.

Response: Thank you very much for your comment. We have made revisions accordingly, however, we have included the names of the statistical tests and p-values in the text, rather than in tables to avoid having too many tables and therefore perplexing readers. Also, p-values were written out when the results were statistically significant, otherwise, we wrote that p was more than 0.05 in order to keep the text as clear as possible.

  1. Tables 5 and 6: Please see the comments on Table 4.

Response: Thank you very much for your comment. We have made revisions accordingly, however, when it comes to the statistical tests and p-values, please refer to the previous answer.

  1. Discussion: Please add any limitations to the study.

Response: Thank you for your comment. We have added the limitation of our study at the end of the first paragraph in the discussion section. The main limitations were a relatively small patient number, a short follow-up period for patients operated on in the last observed period and the inclusion of different sellar pathologies with no distinction.

  1. Out of 77 references, only 11 belongs to the last five years (2019-2023), therefore it is necessary to update the references. No reference belongs to the year 2024. This link https://pubmed.ncbi.nlm.nih.gov/?term=Endoscopic+endonasal+excision+of+pituitary+adenoma&filter=years.2024-2024 might be useful for your study.

Response: Thank you for your comment. We added 7 relevant references published in the last 5 years with 4 published in 2024.

The additional references are:

Ordóñez-Rubiano EG, Capacho-Delgado YA, Jacomussi-Alzate L, Galvis-Oñate KA, Pérez-Chadid D, Tamara-Prieto JA, et al. Dando forma a la curva desde el abordaje transesfenoidal microscópico al endonasal endoscópico para la región selar. Cir Cir. 2024 Jun 12;92(3):11497.

Wong CE, Chen PW, Hsu HJ, Cheng SY, Fan CC, Chen YC, et al. Collaborative Human–Computer Vision Operative Video Analysis Algorithm for Analyzing Surgical Fluency and Surgical Interruptions in Endonasal Endoscopic Pituitary Surgery: Cohort Study. J Med Internet Res. 2024 Jul 4;26:e56127.

Emanuelli E, Zanotti C, Munari S, Baldovin M, Schiavo G, Denaro L. Sellar and parasellar lesions: multidisciplinary management. Acta Otorhinolaryngol Ital. 2021 Apr;41(Suppl. 1):S30–41.

Lubomirsky B, Jenner ZB, Jude MB, Shahlaie K, Assadsangabi R, Ivanovic V. Sellar, suprasellar, and parasellar masses: Imaging features and neurosurgical approaches. Neuroradiol J. 2022;35(3):269–83.

Wang M, Cai Y, Jiang Y, Peng Y. Risk factors impacting intra- and postoperative cerebrospinal fluid rhinorrhea on the endoscopic treatment of pituitary adenomas: A retrospective study of 250 patients. Medicine (Baltimore). 2021;100(49):e27781.

Sitoci-Ficici KH, Sippl C, Prajsnar A, Saffour S, Linsler S. Sellar metastasis: A rare intraoperative finding – surgical treatment, strategies and outcome. Clin Neurol Neurosurg. 2024;241:108280.

Ding Z, Lu X, Wang Q, Qian X, Lu H, Xu R, et al. Endoscopic endonasal surgery of Rathke’s cleft cysts– preoperative imaging evaluation, personalized removal and multilevel sellar floor reconstruction. Clin Neurol Neurosurg. 2024;236:108111. 

Reviewer 2 Report

Comments and Suggestions for Authors

Authors report their experience with a collaboration between neurosurgeons and ENT surgeons in endoscopic endonasal surgery at the Maribor University Medical Center (MUMC)in Slovenia and the University 64 of Pittsburgh Medical Center (UPMC), consisting in visits to the skull base centre at UPMC, consultations and an annual endoscopic skull base surgery cours, and finally a telementoring programme started in 2013 and finished in 2015.  By doing so they review indications and outcomes associated to endoscopic endonasal surgery in the time period from 2011 to 2023.  This is an interesting experience demonstrating how to improve a surgical technique with the collaboration of more experienced colleagues.  However, it would be more interesting to read practical tips and tricks of this collaboration, such as acquisition of instruments, surgical phases, any complications during which tele mentoring was helpful , instead of a general analysis of the surgical series which adds no relevant insight to the pertinent literature. 

Author Response

Response to Reviewer 2

Authors report their experience with a collaboration between neurosurgeons and ENT surgeons in endoscopic endonasal surgery at the Maribor University Medical Center (MUMC)in Slovenia and the University 64 of Pittsburgh Medical Center (UPMC), consisting in visits to the skull base centre at UPMC, consultations and an annual endoscopic skull base surgery cours, and finally a telementoring programme started in 2013 and finished in 2015.  By doing so they review indications and outcomes associated to endoscopic endonasal surgery in the time period from 2011 to 2023.  This is an interesting experience demonstrating how to improve a surgical technique with the collaboration of more experienced colleagues. 

However, it would be more interesting to read practical tips and tricks of this collaboration, such as acquisition of instruments, surgical phases, any complications during which tele mentoring was helpful, instead of a general analysis of the surgical series which adds no relevant insight to the pertinent literature.

Response: Thank you very much for your insightful comments and suggestions. We made corrections accordingly. Several alterations were made in the Methods section. The paragraph about the division of the observed period was moved from the Results into Methods, combined with the number of included patients. Secondly, we disclosed that surgical instrumentation was standardised and that applied surgical techniques were acquired at the UPMC Center for Cranial Base Surgery. We also wrote the name of the equipment used and required for telementoring (Polycom Video Conferencing Equipment, Karl Storz Image 1 and Aida Video System) in case any reader would perhaps like to start the programme at their institution. Due to the time difference between MUMC and UPMC, the times of surgeries had to be adjusted, which also has to be taken into account with telementoring, which was, in our programme, mostly required in periods of resection and reconstruction, that was performed with a nasoseptal flap. Furthermore, we added an description of the setup at MUMC, the start of surgery and our initial approach and the start of telementoring with emphasis on the role of telementoring on surgery (e.g. extended approach, recognition of tumor boundaries and normal pituitary tissue). We also talked about the use of a neuronavigation system and two aspirators; one with a neuronavigated tip, which was extremely helpful. Emphasis was also put on the importance of consent and liability in cases of telementoring.

In the Discussion section, we added a paragraph, which points out the potential drawbacks and limitations of telementoring, which must be addressed prior to the start of any telementoring programme. It is vital that groups involved in telementoring are familiar with their capabilities and dexterity and mutual trust is of paramount importance. Additionally, the question of liability must be cleared before the surgery and in our case, the team at MUMC was solely responsible for the management of patients as well as potential complications. Also, it is crucial that the team operating has a surgical plan laid out before the surgery as well as a backup plan in case of problems with audio-visual connection.

Reviewer 3 Report

Comments and Suggestions for Authors

This article presents a well-conducted study on the impact of surgical telementoring in endoscopic endonasal surgery for skull base pathologies, particularly pituitary adenomas. The retrospective analysis of 94 patients over a 12-year period provides valuable insights into the efficacy of telementoring in enhancing surgical outcomes in a low-volume center. The results indicate a significant increase in complication-free patients and notable improvements in patients' vision and endocrine function, highlighting the potential of telementoring to bridge expertise gaps in less experienced centers.

However, there are notable areas that require attention. Firstly, the article lacks clarity in some sections, particularly in the Methods and Results sections, where the statistical analysis and presentation of data could be more robust and detailed. Additionally, while the study highlights the benefits of telementoring, it does not adequately address the potential limitations and challenges, such as the dependence on reliable technology and potential issues with internet connectivity, which are critical for the success of remote mentoring. Furthermore, the discussion could benefit from a more thorough comparison with existing literature to contextualize the findings within the broader field of endoscopic skull base surgery.

Despite these limitations, the study contributes valuable insights into the potential of telementoring to enhance surgical outcomes in low-volume centers, suggesting that with careful implementation, telementoring can substantially improve outcomes in centers with limited experience in complex surgical procedures.

Comments on the Quality of English Language

Needs minor edit

Author Response

Response to Reviewer 3

This article presents a well-conducted study on the impact of surgical telementoring in endoscopic endonasal surgery for skull base pathologies, particularly pituitary adenomas. The retrospective analysis of 94 patients over a 12-year period provides valuable insights into the efficacy of telementoring in enhancing surgical outcomes in a low-volume center. The results indicate a significant increase in complication-free patients and notable improvements in patients' vision and endocrine function, highlighting the potential of telementoring to bridge expertise gaps in less experienced centers.

However, there are notable areas that require attention.

Firstly, the article lacks clarity in some sections, particularly in the Methods and Results sections, where the statistical analysis and presentation of data could be more robust and detailed.

Response: Thank you very much for your comment. We have made appropriate corrections. We moved the paragraph about the division of the observed time from the Results section to Methods to make the text easier to follow and more transparent. In the Results, we further emphasised the time frame of the telementoring and how surgeries were performed before and after to give readers more insight into the development of the skull base time in a low-volume centre.  Additionally, we have made corrections to Tables; each Table shows the total number of patients included and all abbreviations have been changed to their full terms. In the text, we added the names of the statistical tests used and their p-values.  To make Results more attractive and eye-catching, we added 3 charts that comprehensibly exemplify the impact of telementoring on the newly established skull base team; the first one demonstrates the decreasing rates of postoperative complications, the second one shows the improving postoperative endocrine function and eyesight and the third one illustrates the raising extent of tumor resection per observed period, respectively.

Additionally, while the study highlights the benefits of telementoring, it does not adequately address the potential limitations and challenges, such as the dependence on reliable technology and potential issues with internet connectivity, which are critical for the success of remote mentoring.

Response: Thank you very much for your comment. We have made corrections accordingly. In the Methods section, we further discussed the process of establishing telementoring and the essential requirements (e.g. video conference equipment). Then in the Discussion, we addressed the potential limitations and drawbacks of telementoring, namely issues with audio or video connection.

Furthermore, the discussion could benefit from a more thorough comparison with existing literature to contextualize the findings within the broader field of endoscopic skull base surgery.

Response: Thank you very much for your comment and suggestion. We have made corrections accordingly. We compared the most common signs and symptoms present in our patient population before surgery to those listed in the literature. Also, we discussed the importance of the endoscopic endonasal approach to sellar pathology at the MUMC, where it represents the main surgical modality. We further discussed the number of patients operated on by surgeons at MUMC before becoming proficient, comparing it with the sparse literature available on the topic. Additionally, we added study limitations at the end of the first paragraph in the Discussion section where we recognised that the study’s main limitations were a relatively small patient number, a short follow-up period for patients operated on in the last observed period and the inclusion of different sellar pathologies with no distinction.

Despite these limitations, the study contributes valuable insights into the potential of telementoring to enhance surgical outcomes in low-volume centers, suggesting that with careful implementation, telementoring can substantially improve outcomes in centers with limited experience in complex surgical procedures.

Reviewer 4 Report

Comments and Suggestions for Authors

The authors retrospectively analyzed the EAA data in their center following a telemedicine program. Establishing a pituitary center is not easy, and the author's center showed a clear progression. 

I had several issues regarding this manuscript.

1. Out of all cases, Ten EEA operations were performed in this telementoring program (line 143). When was this telementoring conducted? Was it all in the early phase?

2. It would be more beneficial for the reader if more detail were added regarding the telementoring, such as obstacles or problems during the program. 

3.  The authors combined all suprasellar pathologies, ranging from adenoma to metastasis. All of the pathologies share different natures. Since most cases were adenoma, I think it would have been better if the authors included them only. 

4. Presenting the tables (table 4,5) in graphs would make this manuscript easier to understand. 

Author Response

Response to Reviewer 4

The authors retrospectively analyzed the EAA data in their center following a telemedicine program. Establishing a pituitary center is not easy, and the author's center showed a clear progression.

I had several issues regarding this manuscript.

  1. Out of all cases, Ten EEA operations were performed in this telementoring program (line 143). When was this telementoring conducted? Was it all in the early phase?

Response: Thank you for your comment. The telementoring programme started in the last year of the first observed period (2011-2014), however, for pituitary neoplasms, it was used in the second period (2015-2017).

  1. It would be more beneficial for the reader if more detail were added regarding the telementoring, such as obstacles or problems during the program.

Response: Thank you for your comment and suggestion. We made changes accordingly. In the Methods section, we added a detailed description of the process of establishing the telementoring between MUMC and UPMC and the requirements. In our patient series, no technical difficulties were encountered, however, in the Discussion we described potential issues and drawbacks of telementoring, which must be addressed before the start of the process.

  1. The authors combined all suprasellar pathologies, ranging from adenoma to metastasis. All of the pathologies share different natures. Since most cases were adenoma, I think it would have been better if the authors included them only.

Response: Thank you for your comment and suggestion. In this article, the authors’ main focus was on the development of a skull base team in a low-volume medical centre with the help of experts by means of telementoring, hence, all patients were included despite their histopathological diagnosis, since we wanted to demonstrate the learning curve in acquiring a surgical technique and lower postoperative complications, making the procedure safer for patients. Presenting the data solely for adenomas might be our future project.

  1. Presenting the tables (table 4,5) in graphs would make this manuscript easier to understand.

Response: Thank you for your comment. We have added charts to demonstrate the data in Tables 4, 5 and 6 in order to make the article more transparent and attractive for readers.

Round 2

Reviewer 2 Report

Comments and Suggestions for Authors

Authors partially improved the manuscript; however its scientific relevance remains quite poor.

Reviewer 4 Report

Comments and Suggestions for Authors

Congratulation for your good work. The authors had addressed all points that I addressed. I don't have any issue at all.